# Evaluation of Biocompatibility of PLA/PHB/TPS Polymer Scaffolds with Different Additives of ATBC and OLA Plasticizers

**DOI:** 10.3390/jfb14080412

**Published:** 2023-08-04

**Authors:** Marianna Trebuňová, Patrícia Petroušková, Alena Findrik Balogová, Gabriela Ižaríková, Peter Horňak, Darina Bačenková, Jana Demeterová, Jozef Živčák

**Affiliations:** 1Department of Biomedical Engineering and Measurement, Faculty of Mechanical Engineering, Technical University of Košice, Letná 9, 042 00 Košice, Slovakia; alena.findrik.balogova@tuke.sk (A.F.B.); gabriela.izarikova@tuke.sk (G.I.); darina.bacenkova@tuke.sk (D.B.); jana.demeterova@tuke.sk (J.D.); jozef.zivcak@tuke.sk (J.Ž.); 2Centre for Experimental and Clinical Regenerative Medicine, University of Veterinary Medicine and Pharmacy in Košice, Komenského 73, 041 81 Košice, Slovakia; patricia.petrouskova@uvlf.sk; 3Institute of Materials and Quality Engineering, Faculty of Materials, Metallurgy and Recycling, Technical University of Košice, Letná 9, 042 00 Košice, Slovakia; peter.hornak@tuke.sk

**Keywords:** biocompatibility, PLA/PHB/TPS polymer scaffolds, plasticizers

## Abstract

One of the blends that is usable for 3D printing while not being toxic to cell cultures is the lactic acid (PLA)/polyhydroxybutyrate (PHB)/thermoplastic starch (TPS) blend. The addition of plasticizers can change the rate of biodegradation and the biological behavior of the material. In order to evaluate the potential of the PLA/PHB/TPS material in combination with additives (plasticizers: acetyl tributyl citrate (ATBC) and oligomeric lactic acid (OLA)), for use in the field of biomedical tissue engineering, we performed a comprehensive in vitro characterization of selected mixture materials. Three types of materials were tested: I: PLA/PHB/TPS + 25% OLA, II: PLA/PHB/TPS + 30% ATBC, and III: PLA/PHB/TPS + 30% OLA. The assessment of the biocompatibility of the materials included cytotoxicity tests, such as monitoring the viability, proliferation and morphology of cells and their deposition on the surface of the materials. The cell line 7F2 osteoblasts (Mus musculus) was used in the experiments. Based on the test results, the significant influence of plasticizers on the material was confirmed, with their specific proportions in the mixtures. PLA/PHB/TPS + 25% OLA was evaluated as the optimal material for biocompatibility with 7F2 osteoblasts. The tested biomaterials have the potential for further investigation with a possible change in the proportion of plasticizers, which can have a fundamental impact on their biological properties.

## 1. Introduction

The biological testing of samples is an important part of assessing the safety and toxicity of various materials and substances. The aim of these tests is to determine the effect that a given material or substance has on biological systems. Prevention against potential biological hazards that could arise from the use of medical devices is imperative before clinical use, but the scope of these hazards is wide and complex. In order to minimize risk, all known and foreseeable dangers are identified, and then their risks are evaluated. Currently, it is necessary to comply with ISO standards, focusing on the implementation of the biological evaluation of materials. The standards cover specific aspects of biological evaluations and related tests. ISO 10993 applies to the evaluation of materials and devices in indirect contact with the body during use [1]. There are a wide variety of assays that are used to biologically test samples, including cytotoxicity, genotoxicity, immunotoxicity and hemocompatibility assays. Typically, tests are performed in a laboratory using cells and tissues that are exposed to the material. Subsequently, various parameters are monitored, such as cell growth, changes in genetic material or the production of various chemical substances that may indicate toxicity. Although the ISO 10993 standard series is the most widely accepted reference for biocompatibility testing worldwide, the requirements for biocompatibility testing differ in some countries (e.g., China, America and Canada) [1,2,3]. The biological testing of samples is important not only for assessing the safety of different materials but also for the development of innovative, non-toxic materials with a higher degree of biocompatibility. It also allows the monitoring of the potential adverse effects of materials and substances already used in the medical environment to minimize risks to patients and staff. Biological testing includes two basic categories of tests used in medicine and scientific research: in vivo and in vitro tests. Both types of tests are used by researchers in the study of diseases, the effects of materials on humans and the development of drugs. Each type of test has its advantages and disadvantages, so their use depends on the specific needs and objectives of the test. Currently, a combination of both types of tests is used to maximize accuracy and reliability and to minimize animal experiments [4]. Before assessing biological safety, it is necessary to know the exact chemical and physical composition of the controlled material. Part of the characterization of the material is information on its chemical composition, which is necessary to identify the potential adverse effects of the individual components. Cytotoxicity can be determined by assessing cell morphology, cell growth and reproduction and by measuring cell activity or cell damage. Various cell lines are available for these tests [2].

Within the study of various natural and artificial materials that are suitable for the reconstruction of large tissue defects, finding a suitable material for sufficient vascularization still remains a challenge. In biomedical tissue engineering, various porous scaffold structures from different materials are therefore evaluated at the biological level. A study by Walthers et al. pointed to the fact that micropores facilitate the growth of blood vessels, and macropores in turn influence cell distribution and cell-to-cell interaction [4]. Currently, the goal is to find the most suitable composite polymer material for the preparation of such scaffolds. A frequently studied material is a biocompatible material based on lactic acid (PLA) and polyhydroxybutyrate (PHB). By adding various additives, the physicochemical properties of the PLA/PHB material change. In our pilot study, we decided to test the PLA/PHB material in combination with thermoplastic starch (TPS), in a ratio of 75:25:20, at the biological level. To the basic composite material, we added the additives oligomeric lactic acid (OLA) and acetyl tributyl citrate (ATBC). The main goal was to determine the biocompatibility of the material at the cellular level and to evaluate the survival of cells on the given materials. Based on the obtained results, where we observed changes in cell proliferation, we would like to evaluate the given materials also with other proportional representations in the future.

Lactic acid material has been studied very intensively in recent decades and is classified as GRAS (generally regarded as safe) [5]. In practice, it is becoming increasingly used due to its economical production and its easy use in 3D printing [6]. PLA can be easily modified with various additives, such as fillers, plasticizers, lubricants and pigments. Shortly after its first synthesis, the material was used only for medical applications as bioresorbable materials (surgical sutures) or implants. In biomedical applications, PLA continues to be used for the production of sutures, but also for stents, screws, pins, nails, supporting materials for cell growth and soft tissue implants [7]. Biodegradable materials are a suitable alternative, providing temporary support and long-term tissue growth [8,9]. PLA is a common choice for such applications due to its long degradation time and non-toxicity.

Polyhydroxybutyrate is a biodegradable polymer. Among the most important properties of PHB, considering its usability, are the degradation properties and decomposability of the material. Due to its high degree of biocompatibility, it is used in the form of surgical implants and matrices and in tissue engineering for the growth and proliferation of cells, as well as for surgical threads, fibers and tissues for wound healing. In the pharmaceutical industry, it is used for the production of microcapsules for pills [10,11]. After synthesis, PHB in a powder form is mixed with additives and converted into granules, which can be further processed and mixed. As mentioned above, PHB, like PLA, is prone to degradation at temperatures close to the melting temperature of the materials in question. Therefore, it is necessary to pay attention to temperature control during processing. Processing procedures significantly affect the mechanical properties and morphology of the polymer [12]. For the purpose of this study, we used TPS, which is composed of maize starch with glycerin and water. Due to its natural occurrence, biodegradability and cost-effectiveness, it is called a promising material for scaffold engineering. Starch is highly hydrophilic due to the hydroxyl groups from D-glucopyranose units, which allow for a sufficient biodegradation rate to be achieved [13].

The preparation of polymer mixtures is a frequently used technique in the industrial sector. In this procedure, it is possible to obtain polymers with improved and adapted chemical, physical and mechanical properties via the appropriate regulation of the mass ratio of individual polymer components. The preparation of a mixture of two polymer materials strictly correlates with their miscibility, i.e., the respective solubility parameters. With uniform parameters, good miscibility of the materials is expected, which also depends on the processing temperature, the mass ratio of the mixtures, molecular weights and crystallinity [14]. In the case of medically applied material, the modification and processing of biopolyesters should be performed in a way that does not change their biocompatibility and biodegradability [3,15,16,17]. Less favorable features of the resulting PLA/PHB mixture are fragility and lower strength, which significantly limit their use. To improve the processing and properties of PLA/PHB, active compounds and various types of additives such as plasticizers are often incorporated into the polymer matrix. Several plasticizers are used for PLA, mainly in concentrations between 10 and 30% by weight, such as glycerol, OLA, poly(ethylene glycol) (PEG), citrate esters and low-molecular-weight additives, such as aromatic compounds, including D-limonene, carvacrol and thymol. PHB plasticization is improved by glycerol, glyceryl triacetate, PEG and ATBC. By incorporating such additives, the transition temperature is lowered, and their workability and flexibility are improved. However, the effect of plasticizers reduces the PLA/PHB polymer chain interaction, thereby reducing its barrier properties [18]. Currently, there is an industrial trend to replace traditional plasticizers with natural plasticizers [14,19]. When using a plasticizer, good miscibility between the polymer and the plasticizer must be achieved. Several studies point to the appropriate use of OLA as a plasticizer in PLA and PHB. Both materials belong to the same family of aliphatic polyesters, which contributes to their good compatibility [20,21,22].

As part of our research, we did not come across a publication in which the authors evaluated the given types of materials: I: PLA/PHB/TPS + 25% OLA, II: PLA/PHB/TPS + 30% ATBC and III: PLA/PHB/TPS + 30% of OLA, suitable for the preparation of scaffolds, in biomedical tissue engineering.

## 2. Materials and Methods

The tested scaffolds were prepared from a mixture in the form of filaments, in which the largest proportion was a combination of materials (PLA/PHB/TPS in a ratio of 75:25:20) and plasticizers (ATBC and OLA). Three types of samples of the studied materials were made: sample I: PLA/PHB/TPS + 25% OLA, sample II: PLA/PHB/TPS + 30% ATBC and sample III: PLA/PHB/TPS + 30% OLA, as seen in Table 1.

### 2.1. Preparation of Filaments and Samples

Filaments were manufactured at the Department of Biomedical Engineering, Technical University in Košice, according to the study by Kohan et al., 2022 [23]. Each type of the compared mixed polymer materials was prepared in the form of granules with a weight of approximately 10 g. In this form, the material was placed in the moisture analyzer Radwag 50/1 (RADWAG, Radom, Poland), where it was dried. The whole process lasted 60 min at 80 °C. This procedure was performed due to the hydrophilic properties of PHB to remove excess water. Material with increased resistance was obtained by drying.

Filaments were created from granulate in a COMPOSER 350 filament maker (3devo, Utrecht, The Netherlands). The complete production of fibers took place in an air-conditioned room with a temperature of 18 °C. In the tube, a nitrided steel screw gradually mixed and moved the material through several heat zones. These four extruder temperature zones included Zone 1: 175 °C, Zone 2: 180 °C, Zone 3: 175 °C and Zone 4: 155 °C. The resulting mixture was then pushed through a nozzle with a circular cross-section, resulting in the desired filament. The extruded material was gradually wound onto a coil and cooled. The result was a fiber with a diameter of 1.75 ± 0.05 mm.

The modeling of the samples was carried out in the Magics program (Materialise, Ghent, Belgium). The samples were designed in a cylindrical shape with a diameter of 8 mm and a height of 2 mm, the thickness of the layer was set at 0.2 mm, and the filling of the scaffolds was 50%. The individual layers of the sample were, each time, rotated by 90° relative to the previous bottom layer. The modeled object was converted to STL format (stereolithographic format), suitable for further modifications. After modeling the sample, it was formatted from STL to G-code, which is supported by the 3D printer and based on which the printer obtains the necessary information to print the scaffold. The formatting and preparation of the sample took place in the KISSlicer program, in which the type of material and printing technology were initially defined. Subsequently, in the “Slice Settings” section, the sample was divided into a certain number of slices representing the individual layers of the print, and the position of the sample on the printer plate was chosen. Finally, a G-code was generated in the program, which was used to print the cylindrical scaffold, and its final production was performed via 3D printing.

### 2.2. 3D Printing of Samples (Scaffolds)

According to the study by Findrik Balogolová et al., 2022 [24], the printing parameters are individually dependent on the type of extruded material. The same G-code was used for all samples, which controlled the printer based on previously defined requirements. The printing parameters were chosen to suit the behavior of each material and to be able to compare them. The nozzle of the 3D printer had a diameter of 0.4 mm. The substrate temperature was set to 70 °C, and the printing speed was set to 20 mm/s. These parameters were the same for all materials. The printing time of one scaffold lasted 2 min and 20 s. The individually set parameters included the temperature of the head and the height of the Z-offset of the nozzle. Below, in Table 2, we list the main parameters of printing.

During printing, 20 pieces of samples I, II and III were created from each material (Figure 1)

The macroscope (LEICA WILD M3Z) image provided us with visual information about the morphology of the printed samples. The sampling was carried out at the Faculty of Materials, Metallurgy and Recycling, Technical University in Košice. As a representative sample, we present sample I (PLA/PHB/TPS + 25% OLA) in Figure 2.

For the material of sample I (PLA/PHB/TPS + 25% OLA), its appearance was visually evaluated in comparison with the desired appearance of the scaffold. The scaffold had a circular diameter, and a grid was visible on its surface, which was evenly distributed over the entire surface. The grid consisted of horizontal and vertical strips of fibers that met to form smaller squares. The fibers approaching the edges increased their diameter, and those in contact with the edge were more melted or partially not connected. The surface of the scaffold was smooth, with larger cracks in some fibers. The end of the last thread was visible on the edge, where the nozzle of the 3D printer finished extruding the material. Its pores were small and placed on a grid at regular intervals, forming smaller chambers.

### 2.3. Sterilization of Samples (Scaffolds)

The evaluated samples (PLA/PHB/TSB scaffold and plasticizer) were tested in a 96-well culture plate and a 24-well culture plate after 3D printing. For each sample type, 10 wells were allocated in the case of a 96-well culture plate and 8 wells in the case of a 24-well culture plate. For testing the scaffolds with co-cultivation with cell lines, the samples were sterilized with heat (70 °C) for 1 h and then with UV radiation in a laminar box for 2 h.

### 2.4. Cell Processing and Cultivation

To determine the biocompatible properties of materials based on PLA/PHB/TPS and the plasticizer, the commercial cell line 7F2 osteoblasts isolated from the bone marrow of Mus musculus (ATCC-CRL-12557; American Type Culture Collection, Manassas, United States) were used. The 7F2 osteoblast line stored in liquid nitrogen at −196 °C was thawed in a standard manner. Thawed cells were centrifuged (10 min/1250 rpm, laboratory temperature). The supernatant containing FBS and cryoprotective medium with dimethylsulfoxide (DMSO) was removed.

The 7F2 osteoblasts were resuspended in 1 mL of the culture medium Dulbecco’s Modified Eagle’s Medium, containing glucose (high-glucose DMEM; Merck Life Science spol. s.r.o, Bratislava, Slovakia), 10% fetal bovine serum (FBS) (Merck Life Science spol. s.r.o, Bratislava, Slovakia), 1% antibiotic/antimycotic (ATB + ATM (penicillin-streptomycin-amphotericin B; Merck Life Science spol. s.r.o, Bratislava, Slovakia) and 1% L-glutamate (Merck Life Science spol. s.r.o, Bratislava, Slovakia). Cells were seeded in a 25 cm^2^ culture flask (T-25; TPP) at a concentration of ~1 × 10^4^ cells. Cells were cultured at 37 °C and an atmosphere of 5% CO_2_. Cells that did not adhere to the surface of the culture flask after 24 h were removed. The metabolized culture medium was changed every two to three days. Cell growth was monitored microscopically (Zeiss Axiovert 200, Zeiss, Göttingen, Germany). Cells were cultured until reaching 80–90% confluence.

After reaching approximately 80–90% confluence, the cells were passaged into a culture bottle with a culture area (75 cm^2^). An enzymatic method of trypsinization was used to separate the cells adhered to the surface of the culture bottle. After washing, 1 mL of 0.25% trypsin EDTA (Merck Life Science spol. s.r.o, Bratislava, Slovakia) was added to the cells. Cells with trypsin were incubated at 37 °C for 3 min, during which the trypsin acted on the cells, causing them to detach from the surface of the culture flask. To inactivate trypsin, FBS (Merck Life Science spol. s.r.o, Bratislava, Slovakia) was added to the cells at a ratio of 1:1 [*v*/*v*] trypsin EDTA:FBS. The cell suspension was transferred to a 15 mL centrifuge tube (TPP) and centrifuged (10 min/1250 rpm, laboratory temperature). The supernatant was removed, and the cell pellet was dissociated in 1 mL of culture medium (high-glucose DMEM, 10% FBS, 1% ATB + ATM, 1% L-glutamate), in which the number of cells was determined.

### 2.5. Cell Growth on Samples (PLA/PHB/TSB Scaffold and Plasticizer)

After reaching 80–90% confluence, the cells were trypsinized according to the above procedure. After trypsinization, the cell pellet was resuspended in 1 mL of culture medium (high-glucose DMEM, 10% FBS, 1% ATB + TM, 1% L-glutamate), and the number of cells was determined using a hemocytometer. These 7F2 osteoblasts were used to monitor their growth on a sample (PLA/PHB/TSB scaffold and plasticizer).

Cells (10,000 cells/well) were added to the sterilized samples (Plastic/PHB/TSB scaffold and plasticizer) in a 24-well culture plate. The cell suspension was placed directly on the sample so that the cells preferentially colonized it and not the area around the biomaterial. An amount of 500 µL of the culture medium of high-glucose DMEM, 10% FBS, 1% ATB + TM and 1% L-glutamate was added to the cells. Cells were cultured at 37 °C and an atmosphere of 5% CO_2_. The culture medium was changed every other day. The growth of 7F2 osteoblasts on the samples was monitored for 14 days. The control was 500 µL of the culture medium without cells. The medium sample represented the negative control. Positive and negative controls were measured in parallel.

### 2.6. Morphology of Cells on Samples (Plastic Scaffold from PLA/PHB/TSB and Plasticizer)

Cell morphology and possible changes in shape due to their growth in the presence of biomaterial were monitored using methyl violet. Staining was performed in duplicate on the sixth day of cell culturing. The culture medium was removed from two wells for each sample, and cells were washed twice with 1x PBS (Sigma-Aldrich, St. Louis, MO, USA). An amount of 100 µL of 0.5% methyl violet (Sigma-Aldrich) was added to the cells, and they were incubated for 15 min at room temperature with constant mixing (5 rpm). After incubation, the dye was removed from the wells, and the cells were washed three times with 1x PBS (Sigma-Aldrich). The shape of the cells was observed using a microscope (inverted microscope Zeiss Axiovert 200, Zeiss).

### 2.7. Viable Cell Count and Test

The proliferation of 7F2 osteoblasts was measured using a standard colorimetric test according to the study by Trebuňová et al. [24]. Growth curves were constructed for 7F2 osteoblast cells on three types of samples, namely sample I (PLA/PHB/TSB + 25% OLA), sample II (PLA/PHB/TSB + 30% ATBC), sample III (PLA/PHB/TSB + 30% OLA) and the control. The metabolic activity of the cells on all three types of samples compared to the control was monitored for 14 days. The number and viability of cells was determined using a hemocytometer with trypan blue (Sigma Aldrich, USA) in a trypan blue to cell suspension ratio of 1:9.

### 2.8. Scanning Electron Microscopy

After 6 days of cultivation tracking the growth of 7F2 osteoblasts on the samples (PLA/PHB/TSB scaffold and plasticizer), we visualized cell colonization via scanning electron microscopy. The culture medium was removed, and the cells were washed twice with 1x PBS (Sigma-Aldrich). Subsequently, cells with biomaterials were fixed with a solution of 2.5% glutaraldehyde (Sigma-Aldrich) in 0.1 M PBS (Sigma-Aldrich) for 2 h at room temperature. After fixation, cells with biomaterials were washed three times with 0.1 M PBS (Sigma-Aldrich) for 5 min each time and with constant agitation (5 rpm). Cells on the samples were post-fixed for 1 h in 1% osmium oxide in 0.1 M PBS (Sigma-Aldrich) at room temperature. Subsequently, the cells were washed three times with 1x PBS (Sigma-Aldrich) for 5 min each time and with constant mixing (5 rpm). The samples were dehydrated for 20 min at room temperature using ethanol (ethanol absolute for analysis EMSURE^®^ ACS, ISO, Reag. Ph Eur; Sigma-Aldrich) with an increasing concentration: 50% ethanol, 70% ethanol, 90% ethanol, 95% ethanol and 100% ethanol. After removing the ethanol, the samples were kept in a dry environment until analysis with scanning electron microscopy (SEM). Samples in a culture medium without cells served as a control (K) to monitor the possible degradation of the material.

The morphology of the surface of the prepared samples, their macrostructure and especially their settlement by cells was observed via scanning electron microscopy (SEM) (microscope JEOL JSM-35 CF, Tokyo, Japan) at an acceleration voltage of 25 kV.

### 2.9. Statistical Analysis

The data are presented as the mean ± SEM (Standard Error of the Mean). Significant differences between groups of means were analyzed via an ANOVA. Statistical significance was assumed at a 0.95 CI (Confidence Interval).

## 3. Results

After thawing, the 7F2 osteoblast cell line was plated on a T-25 culture flask at a concentration of ~1 × 10^4^ cells (Figure 3). The non-adherent cells after seeding initially had a typical round to oval shape (Figure 3A), and after 1 h, the adherence of a larger part of the cell population on the surface of the culture bottle and the formation of cell protrusions were observed (Figure 3B).

On the third day of cultivation, the cells in the culture flask reached 60% confluence. Large cells with an irregular stellate shape with various long protrusions were observed, between which intercellular connections were formed (Figure 4).

On the eighth day of culturing, the cells formed a confluent layer (>95%) with multiple cell junctions (Figure 5).

After passaging, 7F2 osteoblast cells, at a number of 10,000 cells per well, were seeded in a 24-well plate (control (pure cells) and samples I (PLA/PHB/TSB + 25% OLA), II (PLA/PHB/TSB + 30% ATBC) and III (PLA/PHB/TSB + 30% OLA). Growth curves representing the activity of 7F2 osteoblast cells were determined via an MTT assay for all three samples compared to the control. The number of cells on monitored samples I, II and III and the control was evaluated after the 5th, 9th, 12th and 14th day. When comparing cell growth on samples I, II and III compared to the control, the smallest decrease in cells occurred in the cells on sample I, but with the increasing number of days, the growth curve of cells on sample III approached the growth curve of cells on sample I. After five days, there was a decrease in the number of cells on all samples, but after nine days, there was an increase in the cells on samples I and III and a decrease in those on sample II. For sample II, the decrease in the number of cells was more prominent than that for cells on samples I and III. From the curves on the graph (Figure 6), we can conclude that sample I (PLA/PHB/TSB + 25% OLA) appeared to be the best for the growth of 7F2 osteoblast cells.

The results of the cell adhesion and cell proliferation of 7F2 osteoblasts are shown in Figure 7. Six independent experiments of cells co-cultured with samples I (PLA/PHB/TSB + 25% OLA), II (PLA/PHB/TSB + 30% ATBC) and III (PLA/PHB/TSB + 30% OLA) and the control (pure 7F2 osteoblast cells) were conducted. Metabolic activity was measured and compared after cell seeding. Figure 7 shows the metabolic activity of the cell population seeded on all three samples compared to the control. The detected differences between the samples were evaluated graphically using box plots and an ANOVA. From the graphs in Figure 7, it is clear that sample I had the largest average values on each measurement day. On the fifth day of measurement, the number of cells was greater in sample II compared to sample III, but on the following days, the number of cells was already greater for sample III. In sample II, the greatest variability of values and also outliers was demonstrated.

Statistically significant differences between sample groups I, II and III were evaluated using an ANOVA, and the result for each day of measurement was *p* < 0.05 (5th day *p* = 0.0000, 9th day *p* = 0.0000, 12th day *p* = 0.0000, 14th day *p* = 0.0000), which means that there was a statistically significant difference between the groups. On the fifth and ninth day, there were statistically significant differences between all observed groups from each other (*p* < 0.05). On the 12th and 14th days, statistically significant differences were confirmed only between groups I and II and between groups II and III (*p* < 0.05). The differences between the groups in samples I and III were not statistically significant (12th day *p* = 0.3714, 14th day *p* = 0.1938).

The growth of 7F2 osteoblast cells on samples I, II and III (scaffolds from biomaterials based on PLA/PHB/TSB and an additive of 25% OLA, 30% ATBC and 30% OLA) and their settlement were monitored on the 1st day, the 8th day and the 11th day (Figure 8, Figure 9 and Figure 10) for 14 days in a 24-well culture plate microscopically. Figure 11 shows the degradation of biomaterials for individual samples I, II and III.

Immediately after the seeding of 7F2 osteoblast cells on individual samples, a typical round shape of non-adherent cells was observed in a density that should be sufficient to populate the biomaterials (Figure 8).

On the eighth day after seeding the cells on the biomaterial, changes in the proliferative activity of the cells between individual samples were observed. The highest proliferation was observed in the case of cells populating sample I (PLA/PHB/TSB + 25% OLA), where elliptical cells with protrusions predominated (Figure 9A). The proliferative activity of cells populating sample III (PLA/PHB/TSB + 30% OLA) was lower compared to that of cells in sample I, with a larger number of dying cells with round and oval shapes without projections (Figure 9C). Minimal cell proliferation and thus colonization of samples by cells was observed in sample II (PLA/PHB/TSB + 30% ATBC), where only dying cells with round and oval shapes without protrusions were observed (Figure 9B).

On the eleventh day after seeding cells on biomaterial, the most significant changes in proliferation were observed. The proliferative activity of the cells on sample I (PLA/PHB/TSB + 25% OLA) decreased slightly, which was related to the observed small number of dying cells (Figure 10A). We also observed a slight decrease in cell proliferation in the case of cells populating sample III (PLA/PHB/TSB + 30% OLA) (Figure 10C). In sample II (PLA/PHB/TSB + 30% ATBC), only dying cells were observed (Figure 10B).

Biomaterials without seeded cells served as a control (K) to monitor possible changes in biomaterials or their degradation. Samples I and III on the 14th day of growth showed minimal degradation changes (Figure 11A,C). The exception was sample II, which showed signs of slight degradation or the release of cells into the medium (Figure 11B).

### 3.1. Monitoring Changes in the Morphology of 7F2 Osteoblasts after Their Growth on the Samples (Plastic Scaffold from PLA/PHB/TSB and Plasticizer)

The growth of cells on biomaterials can be associated with changes in their morphology, by which they adapt to the surface of the used biomaterial. Alternatively, changes in the shape of osteoblasts after their growth on biomaterials were monitored using methyl violet, for microscopic and morphological evaluation. As a representative example, we present Figure 12, with osteoblasts growing in a culture plate without the presence of samples. They were used as a positive control to evaluate changes in the shape of osteoblasts growing on samples I, II and III.

The cells showed an irregular star shape of various sizes with numerous protrusions and clearly visible oval nuclei.

On the 11th day, the highest representation of irregular star-shaped cells with protrusions, which were similar to the shape of osteoblasts growing on culture plastic, was observed in the case of cells cultured in the presence of sample I (PLA/PHB/TSB + 25% OLA). To a lesser extent, round and oval-shaped cells were represented (Figure 13A). Cells populating sample II (PLA/PHB/TSB + 30% ATBC) showed the greatest difference in shape compared to the positive control. We observed a large representation of small dying cells with oval and round shapes (Figure 13B). Cells growing on sample III (PLA/PHB/TSB + 30% OLA) showed the greatest variability in shape. Irregular star-shaped cells with protrusions, elliptical-shaped cells and dying oval and round-shaped cells were also present to a greater extent (Figure 13C).

### 3.2. Evaluation with Electron Microscopy

After 14 days of cultivation monitoring the growth of 7F2 osteoblasts on samples I (PLA/PHB/TSB + 25% OLA), II (PLA/PHB/TSB + 30% ATBC) and III (PLA/PHB/TSB + 30% OLA), the cells on individual samples were evaluated via scanning electron microscopy. In sample I (PLA/PHB/TSB + 25% OLA), a significantly higher number of cells attached to the material was observed in a relatively high density (Figure 14A). In the case of sample II (PLA/PHB/TSB + 30% ATBC) and sample III (PLA/PHB/TSB + 30% OLA), there was a minimal number of cells on the surface of the material, with a low density (Figure 14B,C).

## 4. Discussion

The observed PLA/PHB/TPS materials in combination with OLA and ATBC plasticizers in three variations were optimally observed in a minimum 14-day experiment in order to draw final conclusions about the degree of their compatibility and cytotoxicity. The results of the experiment present knowledge about their suitability for clinical use in the human body. A similar topic was addressed by Kohan et al. [23], who performed cytotoxicity and cell proliferation tests on PLA/PHB + HA/TCP polymers using L929 mouse fibroblasts. The results showed that the materials are not cytotoxic, and the metabolic activity of the cells is greater than 70% in all cases. These results are complemented by another study with a similar composition of materials as that in our experiment, by Čulenová et al. [13], where the results of the MTT test (after 24, 48 and 72 h) pointed to the fact that the material composition of the sample PLA/PHB (60:40) + TPS (30%) had no significant effect on cell proliferation. A slight decrease in the proliferation level during 48 h of culturing was attributed to their adaptation process. The authors evaluated in their study that the material, especially TPS, did not have a toxic effect on the cell culture. Manikandan et al. [25] performed cytotoxicity testing on biopolymers based on lactic and glycolic acid (PLGA) and determined the effect of the composition of these biopolymers on cytotoxicity. The authors used cells cultured in vitro and compared the cytotoxicity of a biopolymer with a higher proportion of glycolic acid to a biopolymer with a higher proportion of lactic acid. The results showed that the biopolymer with a higher proportion of glycolic acid had lower cytotoxicity than that of the biopolymer with a higher proportion of lactic acid. These results are important for the application of these biopolymers in regenerative medicine and tissue engineering, as cytotoxicity can be a critical factor in evaluating the safety and efficacy of these materials. In a study by Moorkoth et al. [26], cytotoxicity was tested using CHO-K1 cells. Testing was performed using PLA and PHB-based materials with additives. The material was exposed to cell cultures, and their growth and survival were subsequently monitored. Cytotoxicity was evaluated based on the number of released enzymes and their effect on the cells. No significant change in viability was observed in any of the tested concentrations, which indicated that the cells were viable, resistant to exposure to the used nanoparticles.

The plasticizers had a significant impact on the mechanical properties of the samples, which greatly influenced their biological properties as well. For example, Arrieta et al. [12] observed the effects of OLA on a PLA/PHB (75:25) blend plasticized with three different concentrations of OLA. OLA improves the crystallization rate of PLA and ensures a better interaction between PLA and PHB. The morphology, structure, thermal stability and mechanical properties of the materials were evaluated. Of the weight of OLA, 20% caused certain fiber defects in the form of spherical pores, which reduced the mechanical properties of the material. The PLA/PHB mixture with OLA at 15% by weight had the best results in the studied areas. In our experiment, also when observing the samples (I (PLA/PHB/TSB + 25% OLA) and III (PLA/PHB/TSB + 30% OLA)), there were defects in the fibers, which could be related to its higher amount in the given mixtures. Nuria Burgos et al. [27] mixed PLA/PHB (85:15) with carvacrol and OLA at 15% by weight of the mixture. The content of these additives positively influenced the permeability and hydrophobic properties of the samples. Armentano et al. [28] addressed the effect of OLA on the processing properties of PLA in PLA/PHB blends, and they confirmed the claim that OLA is one of the most promising options for improving the properties of PLA without compromising biodegradation. They confirmed their experimental results and [28] found that the addition of OLA to PLA/PHB blends significantly reduces their *Tg* values, which confirmed the plasticizing effect of OLA. Similarly, different ratios of OLA were added to the PLA/PHB blend. At a content of 15% and 20% by weight, OLA did not cause a significant change in thermal stability. By increasing its content to 30% by weight, the temperature stability decreased, but the proportional elongation increased up to 370%. We observed changes in temperature stability in our experiment when observing the samples via SEM analysis. Likewise, at 30% OLA content, the temperature stability of sample III (PLA/PHB/TSB + 30% OLA) decreased compared to that of sample I (PLA/PHB/TSB + 25% OLA).

Findrik Balogová et al. [29] were engaged in studying the cytotoxicity of PLA/PHB materials using the additives ATBC and TAC. When monitoring cell growth, ATBC (20% by weight) in PLA/PHB (85:15) and PLA/PHB (50:50) blends had an inhibitory effect on cell growth. The release of oil droplets into the substrate was also observed. The conclusion of the presented experiment also correlates with the mentioned results, where, in the case of sample II (PLA/PHB/TSB + 30% ATBC), the most noticeable decrease in cell proliferation and viability was observed. The components that were released from sample II had an impact on the given properties of the cells, their morphology and settlement on the sample. Burgos et al. [20] studied the properties of PLA/PHB using different plasticizers, including ATBC and OLA. ATBC added at 15% by weight to a PLA/PHB (75:25) blend achieved a significant increase in ductility compared to other plasticizers at this ratio, achieving up to 180% elongation. At the PLA/PHB ratio of 85:15, a larger percentage of plasticizer was used, first with OLA at 15% by weight and later specifically with OLA at 30% by weight. In the mixture with OLA at 15% by weight, the ductility result was very limited, and at a higher OLA content at 30% by weight, a highly elastic material was formed, with a relative elongation of about 350%. Samples I (PLA/PHB/TSB + 25% OLA) and III (PLA/PHB/TSB + 30% OLA) in the presented experiment showed optimal elasticity during 3D printing, which follows their increased ratio of OLA in the mixture. Arrieta et al. [30] studied the mutual effects of PLA and PHB in their blends. PHB at a ratio of 25% by weight increased hydrophobicity, and PLA reduced the brittleness of PHB and improved its workability. However, the PLA/PHB blends still remained brittle, and plasticizers had to be added to the blend to improve processability and tensile properties. The research showed that ATBC (15% by weight) was designed as one of the most effective plasticizers for PLA, PHB and PLA/PHB blends, which was one of the reasons for its use in our study.

With the help of SEM, it is also possible to observe a more detailed morphology of the cells inhabiting the material. A visual representation of the cell population of the biomaterials captured a significantly increased number of cells in the population of sample I (PLA/PHB/TSB + 25% OLA). In the case of samples II (PLA/PHB/TSB + 30% ATBC) and III (PLA/PHB/TSB + 30% OLA), only a minimal number of cells was observable on the surface. SEM also provided us with information about the morphology of individual scaffolds. The morphology of the sample was closest to the 3D STL model in the case of sample I (PLA/PHB/TSB + 25% OLA). In the case of sample II (PLA/PHB/TSB + 30% ATBC), we observed shape variability and smaller spherical pores in the structure, which are likely related to the insufficient miscibility of PLA/PHB with the given plasticizer or its high ratio in the mixture. The most relevant changes occurred in the case of sample I (PLA/PHB/TSB + 25% OLA), in which the PHB phase is difficult to distinguish from the PLA matrix, which indicates a uniform distribution of the components in the composite. Sample III (PLA/PHB/TSB + 30% OLA), from the SEM analysis, showed the presence of fiber-like structures on the surface of the sample, which indicates that part of the molten PLA pellets deformed into fiber-like structures inside the PHB matrix. D’Anna et al. [31] dealt with the microscopic analysis of SEM photographs of PLA/PHB materials and their additives. They investigated mixtures of PLA and PHB materials at different mass ratios, which helped us in the analysis of SEM images. Similarly, Kohan et al. [23] observed the distribution of the ceramic component in the PLA/PHB composite using SEM analysis. Through SEM, Čulenová et al. [13] observed the surface of the PLA/PHB/TPS material and the visible concentration of the cell distribution. They located the largest concentration of cells in the middle of the scaffold, and their morphology was similar to that of fibroblasts. The rougher surface of the samples was attributed to the presence of TPS in the mixture. From the results, they concluded that the mixture of this material is biocompatible and has potential for applications in tissue engineering. Mosnáčková et al. [15] used SEM to visualize changes in the morphology of cells and their possible damage.

The parameters of 3D printing are equally important aspects in the scaffold production process. The quality of the filament is largely dependent on its composition, temperature, extrusion speed and other parameters. Because in our experiment the printing speed and air pressure were set to the same value for all materials, it was assumed that the fiber diameter depended only on the viscosity of the material. In general, viscosity is expected to decrease with temperature, which has been confirmed experimentally. For sample II (PLA/PHB/TSB + 30% ATBC), the nozzle temperature was set to 188 °C, and samples I (PLA/PHB/TSB + 25% OLA) and III (PLA/PHB/TSB + 30% OLA) were at a temperature of 183 °C. At a lower temperature, the layers of the applied biomaterial in sample III (PLA/PHB/TSB + 30% OLA) separated and were not connected to each other. In the work of Kovalčík et al. [32], PHB, PHBV and PHBH were investigated in order to determine their suitability for use in FDM additive manufacturing. The results were that PHB and PHBV are not suitable for FDM due to massive viscosity drops and molecular weight losses. On the other hand, the thermal stability of PHBH was comparable to that of PLA. The FDM-fabricated PHBH scaffolds had excellent mechanical properties, were non-cytotoxic and supported the large proliferation of mouse embryonic fibroblast cells for 96 h. The hydrolytic degradation of PHBH and PLA scaffolds tested in synthetic gastric juice for 52 days confirmed the faster degradation of PHBH than that of PLA. These results suggest that PHBH could be used to fabricate scaffolds with FDM with applications in tissue engineering.

The above studies were performed on materials with a similar composition to that of the samples of our experiment. Testing the biocompatibility of the mixture of polymeric, resorbable materials PLA/PHB/TSB using the OLA plasticizer in two proportions and PLA/PHB/TSB using the ATBC plasticizer is an innovative topic due to the unique composition of the materials and plasticizers used. Based on the results, the given materials are considered suitable for further testing and possible applications in tissue engineering and implantology. The samples showed, depending on their composition, different minimum levels of toxicity and biocompatibility. Plasticizers and other additives had a significant effect on the biological properties of the PLA/PHB materials. In conclusion, we achieved comparable results for the impact of biomaterials on the osteoblast cell line.

## 5. Conclusions

In our study, three types of samples were evaluated, which contained the base material PLA/PHB/TPS at a ratio of 75:25:20 and the ATBC and OLA plasticizers, with three types of samples of the studied materials: sample I (PLA/PHB/TPS + 25% OLA), sample II (PLA/PHB/TPS + 30% ATBC) and sample III (PLA/PHB/TPS + 30% OLA). We observed their effects on the 7F2 osteoblast cell line. Testing was preceded by the production of filaments of individual materials I, II and III and modeling samples in the Magics program.

Growth curves recording the proliferative activity of 7F2 osteoblast cells were followed for all three samples compared to the control. When comparing cell proliferation on samples I, II and III to the control, a minimal decrease in cell proliferation occurred in the observed sample I, but with the increasing number of days, the growth curve of the cells in sample III approached the growth curve of cells on sample I. After five days, there was a decrease in the number of cells on all samples, but after nine days, there was an increase in cells on samples I and III and a decline in cells on sample II. For sample II, the decrease in the number of cells was more prominent than that of cells on samples I and III. From this, we conclude that sample I (PLA/PHB/TSB + 25% OLA) appears to be the best for the growth of 7F2 osteoblast cells.

To monitor the survival of 7F2 osteoblast cells during their growth on samples I, II and III, methyl violet reagent was added to the culture medium and cell suspension in the presence of the material. The results show information about the differences between the given materials. The composition of sample I (PLA/PHB/TPS + 25% OLA) had no significant inhibitory effect on cell proliferation. It decreased only slightly over the days, which was shown by the minimal number of dying cells. We found that the tested material in batch I (PLA/PHB/TPS + 25% OLA) was non-toxic and biocompatible for cell culturing. The material in sample III (PLA/PHB/TPS + 30% OLA) showed a greater decrease in proliferation and a greater number of dying cells than those in sample II (PLA/PHB/TPS + 30% ATBC). In sample II (PLA/PHB/TPS + 30% ATBC), only dying cells and minimal proliferation were observed. The faster death of cells and their lowest proliferation in the case of the material in sample II may be related to its slight degradation and the release of its components into the medium, which was not observed in the materials in samples I and III. The release of the components is likely related to the composition of the material in sample II, due to the different type of plasticizer (ATBC). As a result, we conclude that the material in sample II does not provide sufficient conditions for cell proliferation and viability. Cell growth is also associated with a change in their natural morphology (irregular star-shaped cells of various sizes with numerous protrusions and clearly visible oval nuclei). The results on the 14th day of cell cultivation on biomaterials confirmed the connection between cell growth and changes in their morphology. In the case of material in samples I (PLA/PHB/TPS + 25% OLA), we observed cells that were most similar to the cells that were cultured without the presence of material. We observed the greatest variability in shape in the case of the material in sample III (PLA/PHB/TPS + 30% OLA), and in the case of the material in sample II (PLA/PHB/TPS + 30% ATBC), there were only small, oval and dying cells. Via scanning microscopy, in the case of the material in sample I (PLA/PHB/TPS + 25% OLA), a significantly higher number of cells inhabiting the material was observed at a relatively high density. In the case of materials in samples II (PLA/PHB/TPS + 30% ATBC) and III (PLA/PHB/TPS + 30% OLA), a minimal number of cells, with a low density, was observed on the surface.

From the results, we conclude that the material in sample I (PLA/PHB/TPS + 25% OLA) showed the most suitable biocompatible properties and is suitable for further investigation within the scope of its application in implantology. We conclude that the optimal properties of the material in sample I are related to the type of the used plasticizer (OLA 25%) and its lower ratio. The material in sample II (PLA/PHB/TPS + 30% ATBC) showed the least optimal properties, namely the highest number of dying cells in the shortest time horizon, as well as the largest structural changes after observation via SEM analysis. The properties that the material in sample II showed in this experiment presented a lower degree of biocompatibility, and we assume that they are related to the use of a different plasticizer in the mixture (ATBC 30%) compared to samples I and III. The study of these materials could continue with possible changes in the proportion of the plasticizer in the mixtures. The results of the biocompatibility tests that the material in sample III (PLA/PHB/TPS + 30% OLA) showed in short-term exposure are acceptable. Each of the given materials has the potential for further study, and with a slight change in their additives, it is possible that they can achieve the desired properties to be part of the human body in the future.

## Figures and Tables

**Figure 1 jfb-14-00412-f001:**
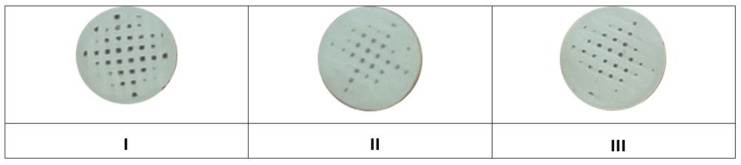
Printed samples of materials (**I**–**III**).

**Figure 2 jfb-14-00412-f002:**
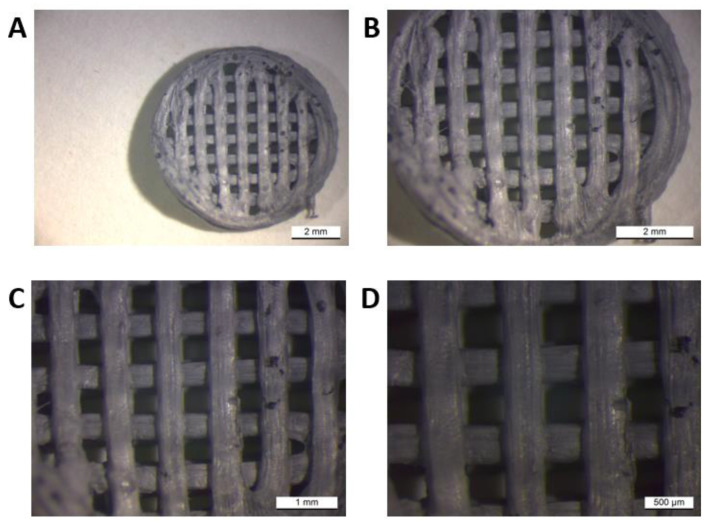
Image of morphology of the printed sample I (PLA/PHB/TPS + 25% OLA) at different magnifications using the macroscope at different resolution ((**A**,**B**): with scale 2 mmm, (**C**): 1 mm, (**D**): 500 μm).

**Figure 3 jfb-14-00412-f003:**
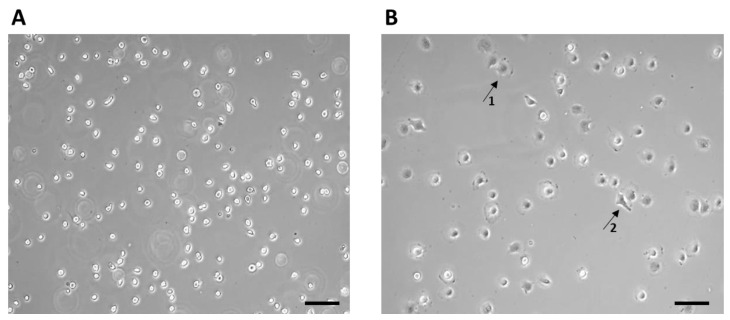
7F2 osteoblast cell line the first day after thawing: (**A**) 7F2 osteoblast cell line directly after plating on a T-25 culture flask. Scale 50 µm. (**B**) 7F2 osteoblast cell line 1 h after plating on a T-25 culture flask. Arrows show an example of a cell adhering to the surface of a culture bottle (1) and the formation of cell protrusions (2). Scale 50 µm.

**Figure 4 jfb-14-00412-f004:**
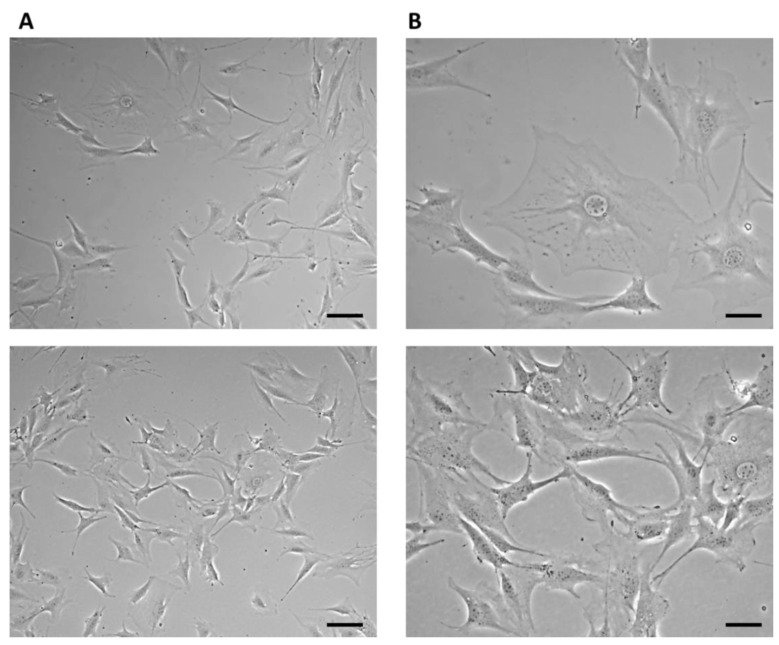
7F2 osteoblast cell line on the third day after thawing. 7F2 osteoblasts acquired an irregular star shape with numerous protrusions forming intercellular contacts. (**A**) Scale bar 50 µm; (**B**) Scale bar 20 µm.

**Figure 5 jfb-14-00412-f005:**
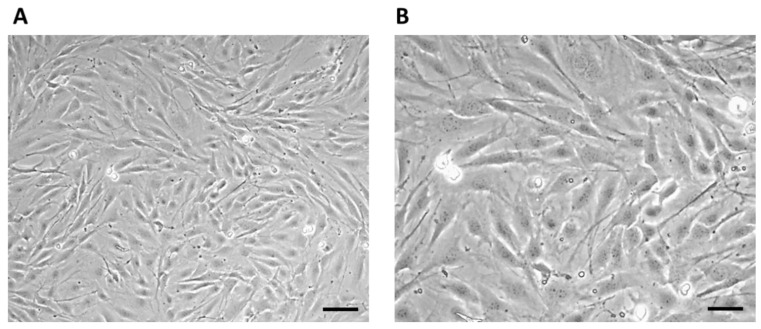
Cell line 7F2 osteoblasts on the eighth day of culturing: (**A**) Scale bar 50 µm; (**B**) Scale bar 20 µm.

**Figure 6 jfb-14-00412-f006:**
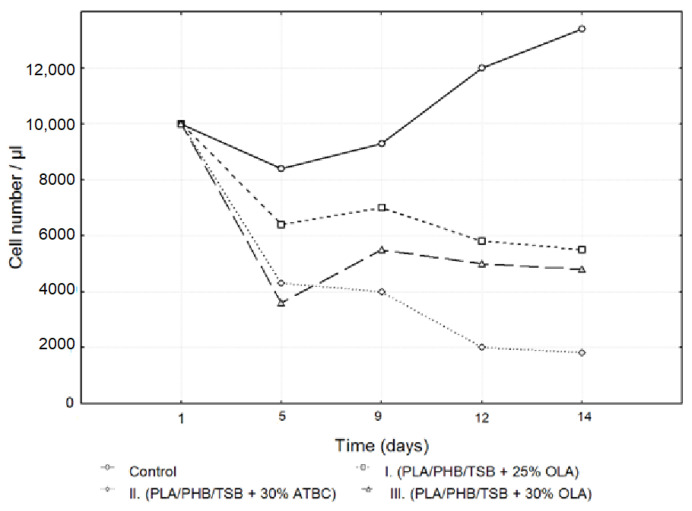
Growth curves of 7F2 osteoblast cells. The data represent the mean ± SEM (Standard Error of the Mean) of six independent experiments.

**Figure 7 jfb-14-00412-f007:**
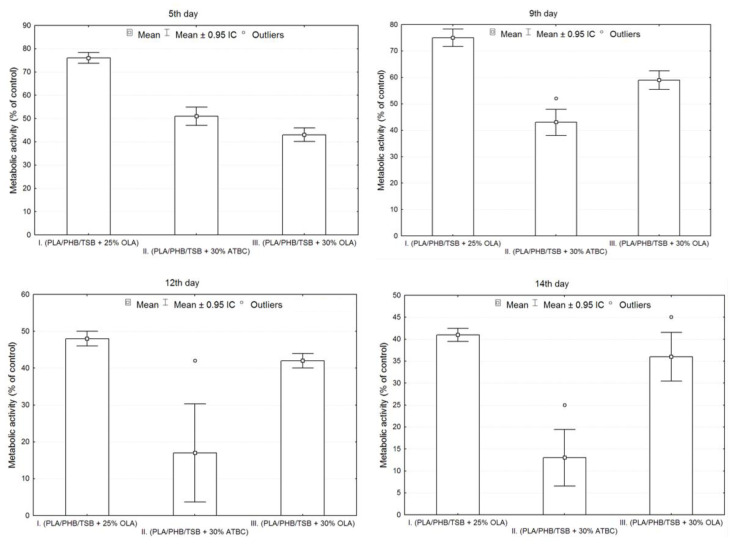
Metabolic activity of 7F2 osteoblast cells co-cultured on samples I, II and III. Bars represent the mean ± 0.95 CI (Confidence Interval) of six independent experiments. Statistical significance was assessed using ANOVA (° < 0.05).

**Figure 8 jfb-14-00412-f008:**
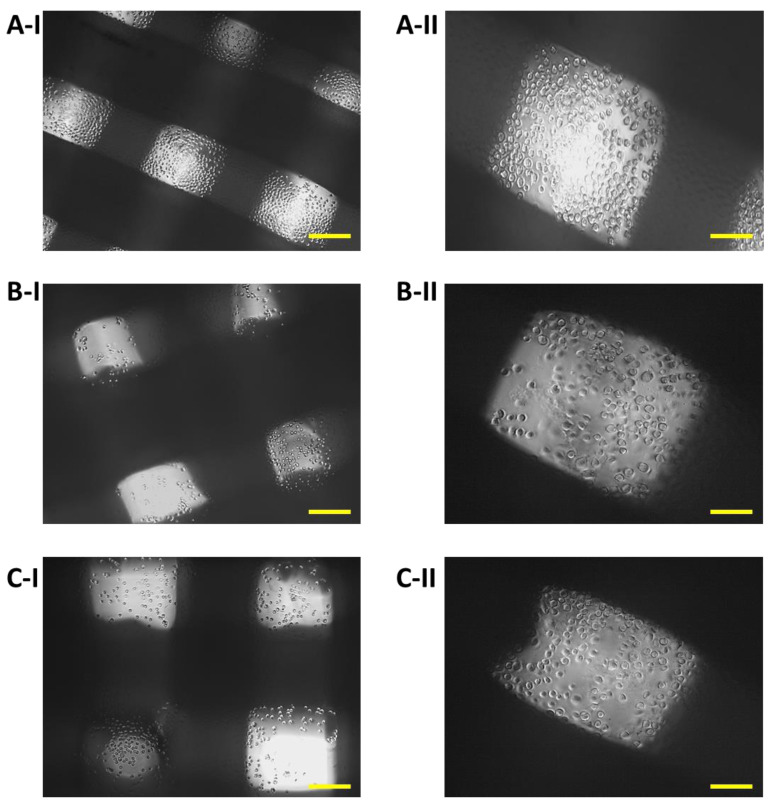
Proliferation of 7F2 osteoblasts on samples the first day after seeding: (**A**) 7F2 osteoblast cells on sample I (PLA/PHB/TSB + 25% OLA); (**B**) 7F2 osteoblast cells on sample II (PLA/PHB/TSB + 30% ATBC); (**C**) 7F2 osteoblast cells on sample III (PLA/PHB/TSB + 30% OLA). Scale bars: I—100 µm; II—50 µm.

**Figure 9 jfb-14-00412-f009:**
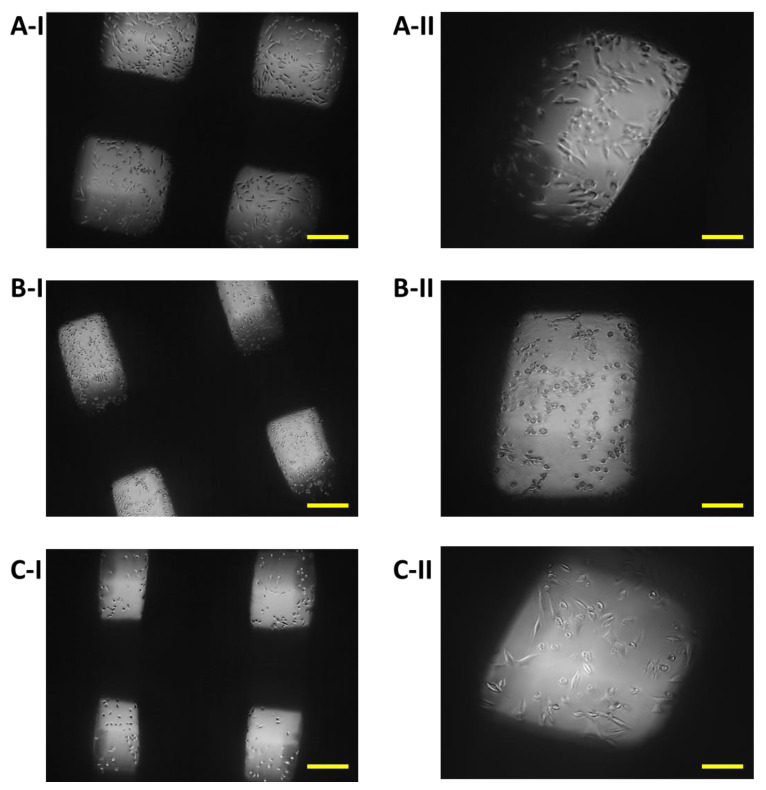
Proliferation of 7F2 osteoblasts on samples on the eighth day after seeding: (**A**) Osteoblasts on sample I (PLA/PHB/TSB + 25% OLA)—best proliferative activity; (**B**) Osteoblasts on sample II (PLA/PHB/TSB + 30% ATBC)—without cell proliferation; (**C**) Osteoblasts on sample III (PLA/PHB/TSB + 30% OLA)—moderate cell proliferation. Scale bars: I 100 µm; II 50 µm.

**Figure 10 jfb-14-00412-f010:**
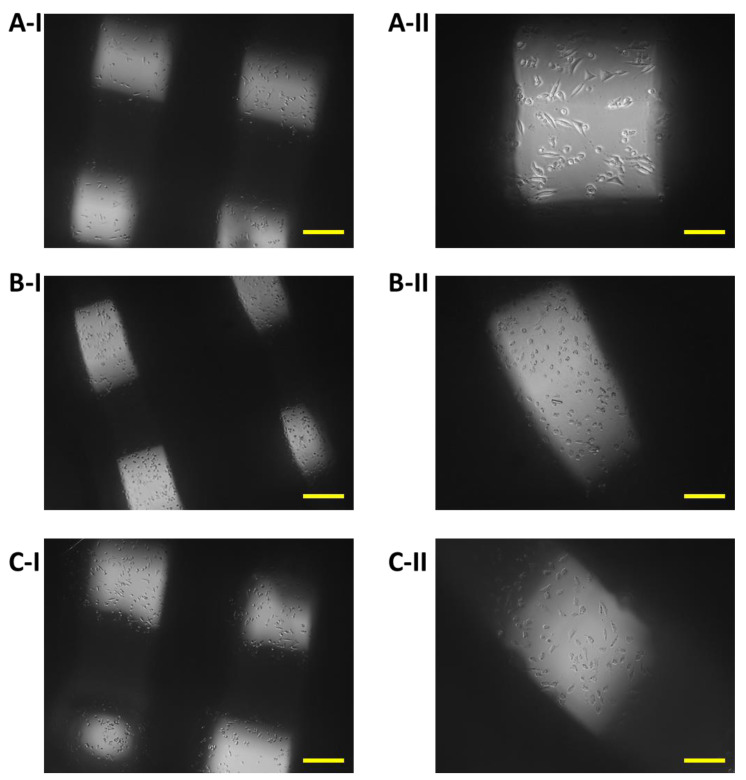
Proliferation of 7F2 osteoblasts on samples on the eleventh day after seeding: (**A**) Osteoblasts on sample I (PLA/PHB/TSB + 25% OLA)—lower rate of cell proliferation; (**B**) Osteoblasts on sample II (PLA/PHB/TSB + 30% ATBC)—no cell proliferation; (**C**) Osteoblasts on sample III (PLA/PHB/TSB + 30% OLA)—lower rate of cell proliferation. Scale bars: I 100 µm; II 50 µm.

**Figure 11 jfb-14-00412-f011:**
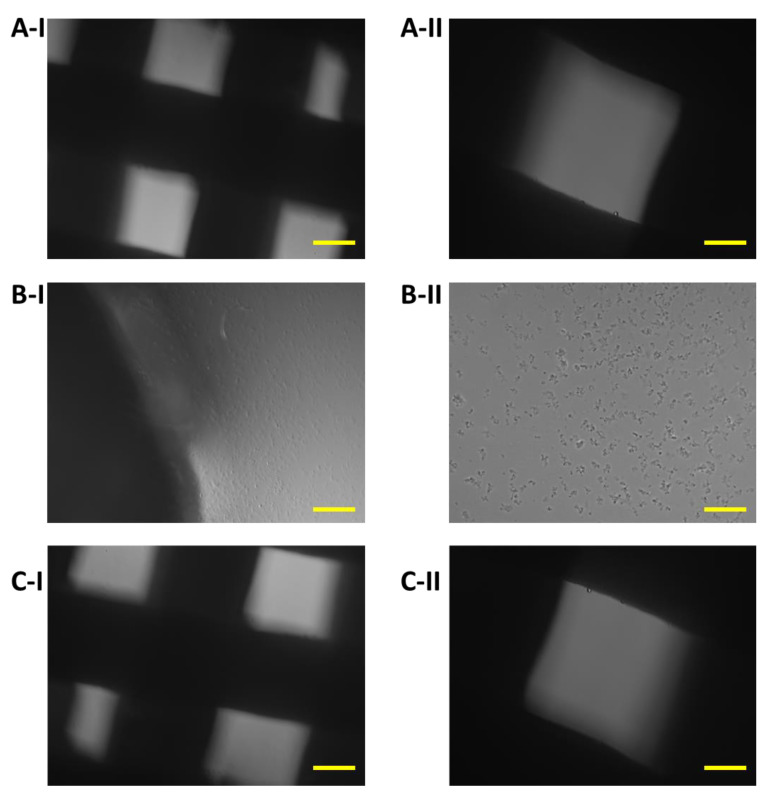
Changes in biomaterial on the fourteenth day in the culture medium: (**A**) Sample I (PLA/PHB/TSB + 25% OLA); (**B**) Sample II (PLA/PHB/TSB + 30% ATBC); (**C**) Sample III (PLA/PHB/TSB + 30% OLA). Scale bars: I 100 µm; II 50 µm. Samples I and III showed no changes, and sample II showed signs of degradation.

**Figure 12 jfb-14-00412-f012:**
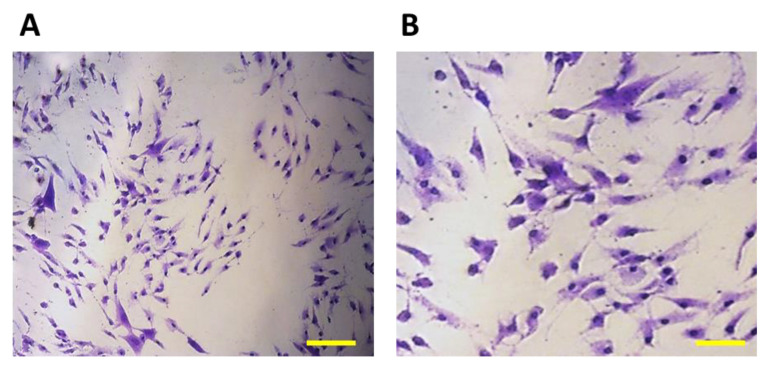
7F2 osteoblasts on the 6th day of their growth. Osteoblasts growing in a 24-well culture plate without the presence of samples represented a positive control for the assessment of osteoblast growth on samples I, II and III. (**A**) 50 µm magnification; (**B**) 20 µm magnification.

**Figure 13 jfb-14-00412-f013:**
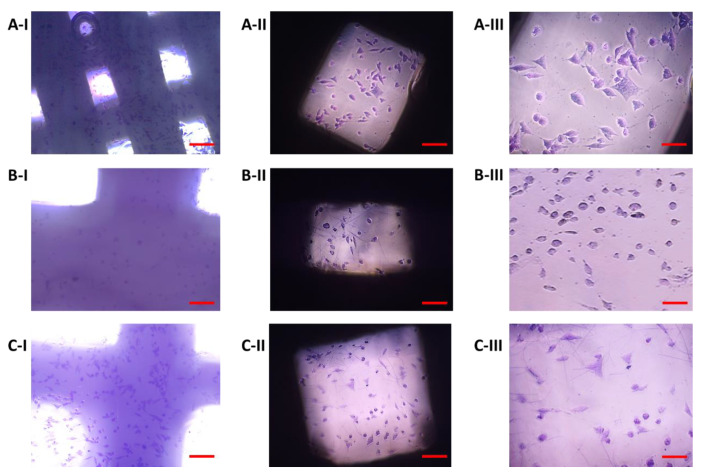
Morphology of 7F2 osteoblasts on day 11 of their growth on samples I (PLA/PHB/TSB + 25% OLA), II (PLA/PHB/TSB + 30% ATBC) and III (PLA/PHB/TSB + 30% OLA): (**A**) Osteoblasts on sample I—irregular star-shaped cells with prominent cell protrusions; (**B**) Osteoblasts on sample II—a small number of oval and round cells; (**C**) Osteoblasts on sample III—irregular star-shaped cells and oval and round cells. Scale bars: I 100 µm; II 50 µm; III 20 µm.

**Figure 14 jfb-14-00412-f014:**
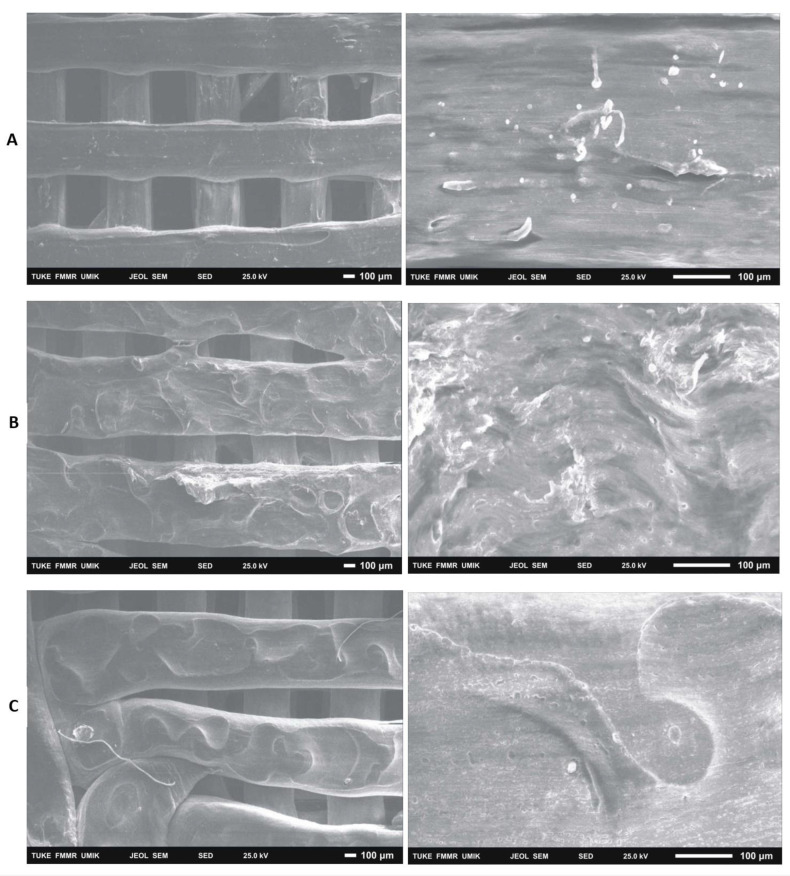
Colonization of samples I (PLA/PHB/TSB + 25% OLA), II (PLA/PHB/TSB + 30% ATBC) and III (PLA/PHB/TSB + 30% OLA) by 7F2 osteoblast cells SEM: (**A**) Sample I—large number of cells on the surface; (**B**) Sample II—low number of cells on the surface of slightly degraded material; (**C**) Sample III—low number of cells.

**Table 1 jfb-14-00412-t001:** Contents of materials 111, 145 and 146.

Samples	PLA (wt.%)	PHB (wt.%)	TPS (wt.%)	Plasticizer (wt.%)
I	75	25	20	25 OLA
II	75	25	20	30 ATBC
III	75	25	20	30 OLA

**Table 2 jfb-14-00412-t002:** Individual printing parameters for specific materials.

Sample	Print Speed (mm/s)	Platform Temperature (°C)	Head Temperature (°C)	Nozzle Diameter (mm)	Z-Offset (mm)	Number of Samples (pcs)
I	20	70	183	0.4	−3	20
II			188		−2.8	
III			183		−2.7	

## Data Availability

There are no additional research data.

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
