# Peer review of "Evaluation of Biocompatibility of PLA/PHB/TPS Polymer Scaffolds with Different Additives of ATBC and OLA Plasticizers"

_jfb, 2023, doi:10.3390/jfb14080412_

Round 1
Reviewer 1 Report
The manuscript is devoted to a topical issue - the development of new biodegradable scaffolds for the manufacture of temporary prostheses. To this end, the authors tested several ternary polymer blends based on lactic acid, polyhydroxybutyrat, and thermoplastic starch with the addition of plasticizers. The choice of a multicomponent composition is partially confirmed by the references to the literature. Cylindrical meshes were obtained by 3D printing as scaffold models. For the evaluation of the biocompatibility of the materials, cytotoxicity tests including monitoring of viability, proliferation, morphology of cells, and their settlement on the surface of the materials were performed. It is shown that the addition of plasticizers significantly affects these parameters. PLA/PHB/ATBC + 25% OLA was chosen as the optimal material.
The bibliography is quite recent and reflects the current state of the problem. The practical significance is apparent.
There are a few remarks to note:
1. The introduction should be completed with a justification of why the authors chose these objects for the research.
2. The abbreviation TPS, which occurs for the first time in line 134, was previously deciphered only in the abstract. It should be deciphered in the text.
3. In line 169 instead of “2.2.3. D printing” should be written “2.2. 3D printing”.
After taking into account these minorcomments, the manuscript may be published.
Author Response
We send the answers to the comments of Reviewer 1 in the attachment.

Reviewer 2 Report
In this manuscript, Marianna Trebuňová et al. analyzed the biocompatibility of a material based on lactic acid in combination with additives. The evaluation of the biocompatibility of the materials included cytotoxicity tests such as the monitoring of viability, proliferation, morphology of cells, and their settlement on the surface of the materials. In my opinion. A Major revision is needed. The details are listed as follows.
1. The authors gave too many details in the Abstract, which should be shortened and rewritten.
2. The authors should highlight more about the novelty and significance of this work.
3. The authors should give a table to compare the biocompatibility of PLA or PHB or TPS polymer-based materials with additives.
4. The authors should check and make the format of references more uniform.
NIL
Author Response
We send the answers to the comments of Reviewer 2 in the attachment.

Reviewer 3 Report
The authors prepared three types of materials : I: PLA/PHB/TPS + 25% OLA, II: PLA/PHB/TPS + 30% ATBC, III: PLA/PHB/TPS + 30% OLA, and evaluated their biocompatibility. I can not recommend accept this manuscript now, major revision is needed.
1. The novelty of this manuscript is low, because as authors mentioned, mixuters of ATBC as additives in PLA, PHB have been reported and the strength properties of them have been investigated(in the last paragraph of Introduction). The authors should revise the introduction section and demonstrate why to do this work.
2. OLA as additives in PLA and PHB should be reviewed in Introductuon.
3. The mixture ratios are too less to support the conclusion, for OLA only two ratio, and for ATBC, only one ratio. How about with more OLA or less OLA? Besides, in Figure 2, why only sample I were tested? Corresponding experiment results showed be added.
4. Some format issues should be revised: the full name of abbreviation (ATBC and OLA ) were given twice; Figure X and Fig. X should be unified; Table 1 and Tab.1.....
5. In Figure 12A, all results of sample I, II, III should be provided, the result maybe different.
Can be improved.
Author Response
We send the answers to the comments of Reviewer 3 in the attachment.

Round 2
Reviewer 2 Report
The authors have addressed most of the comments and revised the manuscript properly. I have no more comments. Finally, I would like to recommend this manuscript to be accepted.
Reviewer 3 Report
This is an research article, not a pilot study. The author do not want to or can not provide additional experiment results, so the conclusion can not be strongerly supported. So I recommend reject this manuscript.
can be improved.